# Diagnostic Accuracy of Liquid Biopsy in Endometrial Cancer

**DOI:** 10.3390/cancers13225731

**Published:** 2021-11-16

**Authors:** Marta Łukasiewicz, Krzysztof Pastuszak, Sylwia Łapińska-Szumczyk, Robert Różański, Sjors G. J. G. In ‘t Veld, Michał Bieńkowski, Tomasz Stokowy, Magdalena Ratajska, Myron G. Best, Thomas Würdinger, Anna J. Żaczek, Anna Supernat, Jacek Jassem

**Affiliations:** 1Laboratory of Translational Oncology, Intercollegiate Faculty of Biotechnology, Medical University of Gdańsk, 80-211 Gdańsk, Poland; marta.lukasiewicz@gumed.edu.pl (M.Ł.); krzpastu@pg.edu.pl (K.P.); azaczek@gumed.edu.pl (A.J.Ż.); 2Department of Algorithms and Systems Modelling, Faculty of Electronics, Telecommunication and Informatics, Gdańsk University of Technology, 80-233 Gdańsk, Poland; 3Department of Gynecology, Gyneacological Oncology and Gynecological Endocrinology, Medical University of Gdańsk, 80-211 Gdańsk, Poland; slapin@wp.pl (S.Ł.-S.); rozanskirobert@gmail.com (R.R.); 4Department of Neurosurgery, Amsterdam University Medical Center, Vrije Universiteit Amsterdam, Cancer Center Amsterdam, 1081 HV Amsterdam, The Netherlands; g.intveld1@amsterdamumc.nl (S.G.J.G.I.V.); m.g.best@amsterdamumc.nl (M.G.B.); t.wurdinger@amsterdamumc.nl (T.W.); 5Brain Tumor Center Amsterdam, Amsterdam University Medical Center, Vrije Universiteit Amsterdam Medical Center, Cancer Center Amsterdam, 1081 HV Amsterdam, The Netherlands; 6Department of Pathomorphology, Medical University of Gdańsk, 80-211 Gdańsk, Poland; michal.bienkowski@gumed.edu.pl; 7Department of Clinical Science, University of Bergen, 7800 Bergen, Norway; tomasz.stokowy@k2.uib.no; 8Centre of Biostatistics and Bioinformatics Analysis, Medical University of Gdańsk, 80-211 Gdańsk, Poland; 9Department of Biology and Medical Genetics, Medical University of Gdańsk, 80-211 Gdańsk, Poland; magda.ratajska@otago.ac.nz; 10Department of Pathology, University of Otago, Dunedin 9016, New Zealand; 11Department of Oncology and Radiotherapy, Medical University of Gdańsk, 80-211 Gdańsk, Poland; jjassem@gumed.edu.pl

**Keywords:** endometrial cancer, tumor educated platelets, circulating tumor DNA, molecular markers, liquid biopsy, artificial intelligence

## Abstract

**Simple Summary:**

The number of endometrial cancer (EC) cases is constantly growing. However, the current diagnostic approach is still rather imprecise, leaving 1/3 of patients temporarily undiagnosed. Moreover, final diagnosis is made after the surgery. That mean the histology of tumor, which influences scope of resection, is uncertain during procedure. This results in over- and undertreatment of EC patients. Those diagnostic problems might be solved by liquid biopsy—a new, minimally invasive method to obtain tumor biomarkers. Therefore, this study aimed to evaluate the usefulness of information obtained from liquid biopsy components (tumor educated platelets and circulating tumor DNA) coupled with random forest algorithm and deep neural networks to diagnose EC patients and evaluate tumor histology preoperatively.

**Abstract:**

Background: Liquid biopsy is a minimally invasive collection of a patient body fluid sample. In oncology, they offer several advantages compared to traditional tissue biopsies. However, the potential of this method in endometrial cancer (EC) remains poorly explored. We studied the utility of tumor educated platelets (TEPs) and circulating tumor DNA (ctDNA) for preoperative EC diagnosis, including histology determination. Methods: TEPs from 295 subjects (53 EC patients, 38 patients with benign gynecologic conditions, and 204 healthy women) were RNA-sequenced. DNA sequencing data were obtained for 519 primary tumor tissues and 16 plasma samples. Artificial intelligence was applied to sample classification. Results: Platelet-dedicated classifier yielded AUC of 97.5% in the test set when discriminating between healthy subjects and cancer patients. However, the discrimination between endometrial cancer and benign gynecologic conditions was more challenging, with AUC of 84.1%. ctDNA-dedicated classifier discriminated primary tumor tissue samples with AUC of 96% and ctDNA blood samples with AUC of 69.8%. Conclusions: Liquid biopsies show potential in EC diagnosis. Both TEPs and ctDNA profiles coupled with artificial intelligence constitute a source of useful information. Further work involving more cases is warranted.

## 1. Introduction

Liquid biopsy, being a minimally invasive alternative to surgical tissue biopsies, has recently revolutionized cancer diagnostics. This type of analysis is typically achieved using a blood sample. The procedure enables interrogation of tumor-derived material such as circulating tumor cells (CTCs), circulating tumor DNA (ctDNA), circulating free RNA (cfRNA), tumor-derived extracellular vesicles (EVs), and more recently, tumor educated platelets (TEPs), all present in the body fluids of cancer patients [1,2]. So far, mutational burden [3] and methylation pattern in ctDNA [4], along with circulating miRNA signatures [5], were used for screening and diagnosis of EC. Liquid biopsies as a diagnostic tool are expected to predict: (a) tumor aggressiveness with ctDNA level determination [6]; (b) relapse with ctDNA mutational profiling [7]; (c) recurrence risk using EV annexin A2 and L1CAM level analysis [8]; and (d) immunotherapy response by assessing microsatellite instability [9].

Liquid biopsies are poorly explored in endometrial cancer (EC). There is an apparent need for improving preoperative diagnosis of EC. Limitations of the currently used diagnostic procedures (insufficient tissue amount, technical failures) leave approximately 30% of EC patients temporarily undiagnosed [10]. This problem is even more important nowadays due to a rising incidence of EC cases among younger women and higher prevalence of more aggressive EC subtypes [11,12].

An unexplored method of liquid biopsy in EC is TEPs technology. TEPs are platelets that interact with cancer cells and are heavily involved in the progression of solid tumors, and their transcriptomic profile changes dramatically under the influence of the disease. Best et al. have demonstrated that the platelet transcriptome allows for distinguishing between healthy, asymptomatic subjects and cancer patients in various malignancies including glioblastoma, sarcoma, non-small cell lung cancer, pancreatic adenocarcinoma, colorectal cancer, and breast cancer [2,13]. To the best of our knowledge, there have been no investigations on platelet transcriptome in EC patients. This approach might find its application in preoperative diagnosis of EC, as the planning of surgical intervention based on traditional biopsy results is relatively imprecise. As mutations found in ECs reflect their histology [14,15], ctDNA analysis could also be applied to preoperative histology evaluation and may optimize treatment planning.

Both TEPs and ctDNA can be obtained from one vial of blood. Thus, we decided to perform both assays in a parallel manner. We hypothesized that the analysis of TEPs could aid initial EC diagnosis by discrimination between healthy women and EC patients, whereas ctDNA interrogation would facilitate EC histology evaluation. As sequencing of either RNA or DNA generates massive, difficult to interpret amounts of data, we used the artificial intelligence for decision making: deep neural network and random forest, as presented in Figure 1.

## 2. Materials and Methods

Sample collection: Liquid biopsies were collected from gynecologic patients treated at the Department of Gynecology, Oncologic Gynecology and Gynecological Endocrinology, Medical University of Gdańsk (MUG, Gdańsk, Poland) between 2017 and 2019. Additionally, the sequencing data were obtained from: (a) platelets collected from 204 healthy women referred as healthy donors (Dutch cohort) [16]; (b) primary tumors of 519 EC patients available at Genomic Data Commons (GDC) Data Portal (referred as GDC cohort). To avoid reproducibility bias, all TEPs samples and all ctDNA samples were sequenced in one institution (Amsterdam and Gdańsk, Poland respectively).

The inclusion criteria for this study included EC before the initiation of any treatment or benign gynecologic condition (BGC), and age above 18 years. Each patient signed an informed consent form. The study was accepted by the Independent Ethics Committee of Medical University of Gdańsk (NKBBN/434/2017). Procedures involving human subjects were in accordance with the Helsinki Declaration of 1975, as revised in 1983. Stage evaluation was based on classification of Federation of Gynecology and Obstetrics (FIGO) from 2014 [17]. EC was divided into type 1 (endometrioid and mucinous histology) and type 2 (serous, clear cell and undifferentiated tumors) [18].

Blood processing: Blood from gynecologic patients was collected into 6 mL BD vacutainer tubes with EDTA as an anti-coagulant. Samples were processed up to 48 h after collection according to the protocol presented in Figure 2. Frozen platelets were shipped on dry ice to VUmc Cancer Center Amsterdam (Amsterdam UMC, the Netherlands) for further processing and sequencing, strictly according to the protocol published by Best et al. [19].

Platelet RNA sequencing and sample classification: Expression data for each sample were normalized using DESeq2 package [20] with variance stabilizing transformation [21] and then annotated with Gencode v19 GRCh37 [22]. Samples with less than 100,000 total reads were excluded from further analysis. For platelet RNA-seq profiles, we used imPlatelet classifier published by Pastuszak et al. [23]. The classifier is based on deep neural networks and yields a TEPs score that can be interpreted as a quantitative measure of the similarity of the considered transcription profile to the profile from a healthy donor (0) and patients with EC or benign gynecologic control (1) from the training cohort. In this classification, we used leave-one-out cross-validation, where in each fold, the test set consisted of one distinct sample. Cross-validated area under curve (AUC) was computed using cvAUC r package [24]. Cross-validated receiver operating characteristic (ROC) curves were generated using ROCIT r package [25].

Germline DNA and ctDNA extraction: Germline DNA was isolated from 100 µL buffy coat using QIAamp^®^ DNA Blood Mini Kit according to the manufacturer’s guidelines. Circulating tumor DNA was isolated from 0.5–3 mL of plasma, using the QIAamp^®^ MinElute ccfDNA Mini Kit according to the manufacturer’s guidelines, with an additional step of double plasma spinning after thawing (10 min at 1600× *g*, followed by 10 min at 16,000× *g*) in order to reduce plasma contamination with germline DNA. Quantification of the extracted ctDNA was performed on TapeStation 4200 platform using the Agilent High Sensitivity D1000 ScreenTape and Agilent High Sensitivity Assay Kit. Quantification of the extracted germline DNA was performed with the use of Qubit Flourometer and Qubit DNA High Sensitivity Assay Kit. Extracted material was stored at −80 °C for later library construction.

Library construction and sequencing: For ctDNA and germline DNA sequencing, library construction was performed using QIAseq Targeted Human Colorectal Cancer Panel covering 71 genes. Complete gene list is available in Appendix A. The panel was composed of 2929 primers, with a size of 215,328 bp. Genes typically differentiating endometrioid from non-endometrioid EC include ERBB2, KRAS, CDH1, PTEN, TP53, PIK3R1, and PIK3CA. However, it is highly likely that some mutations in ctDNA of EC patients were below the limit of detection and thus remain uncalled [26]. This prompted us to apply machine learning instead of manual (subjective) EC-type classification. For the library construction, we used 0.5–13 ng of ctDNA and 40 ng germline DNA. The procedure was performed according to the manufacturer’s guidelines. Prepared libraries were stored at −20 °C. Libraries were sequenced in a paired-end (2 × 151 bp) manner on HiSeq X Ten (ctDNA libraries) and MiniSeq (germline libraries) Illumina platform. Expected coverage for ctDNA-based libraries was 5000×, whereas for germline DNA-based libraries, it was 100×.

Tumor DNA sequencing and sample classification: For DNA sequencing, the alignment was performed using bwa-mem [27] 0.7.17-r1188, included in the cgp-wxs docker container version 3.1.7, suitable for exome and panel sequencing. Reads were aligned to the reference human genome GRCh37d5, with filtering of the capture kit provided by the panel manufacturer (DHS-002Z, Human Colorectal Cancer Panel). Mosdepth [28] tool (mosdepth 0.2.4 docker container) was applied to evaluate the quality and depth of the sequenced samples. Finally, single nucleotide variants were called using sinvict [29] version 1.0. Determination of EC type based on ctDNA mutational profile was performed using a random forest classifier. The model development procedure involved DNA sequencing data from 519 primary tumors. The procedure and results of the classifier development are presented as Appendix A and Appendix A. The model development procedure was tested on two independent cohorts. The classifier divides the patients into two groups: patients who have a tumor with endometrioid (type 1) and non-endometrioid histology (type 2).

## 3. Results

### 3.1. Patients

The study group included liquid biopsy samples collected from 53 EC patients (16 ctDNA and 37 TEPs samples, not matched), 38 patients with benign gynecologic conditions (TEPs) and 204 healthy donors (TEPs). Benign gynecologic conditions included myomas, endometriosis, cysts, polyps, and Brenner tumors. Primary tumor samples were only used for the training of the classifier. Patient characteristics are summarized in Table 1, and a more detailed sample list in shown in Appendix A.

### 3.2. Platelet-Based Classification

In EC discrimination versus non-cancer patients and healthy volunteers, imPlatelet classifier reached an accuracy of 99.7%, 93.1% and 93.1% in the training (*N* = 168), validation (*N* = 111) and test set (*N* = 279, LOOCV), respectively (Figure 3). However, average TEPs score of classification was 80.1% for EC patients, 50.0% for patients with benign gynecologic condition, and 0.01% for healthy donors in the test set (Figure 4).

### 3.3. ctDNA-Based Classification

The liquid biopsy cohort applied to the trained random forest classifier included the ctDNA mutations from the patients with endometrioid and non-endometrioid EC collected in Gdańsk. On the test set, which included mutational profiles found in primary tumors (data from GDC database, not matched with the liquid biopsy samples), the classifier reached an AUC of 96%. Classification of plasma ctDNA, using the optimal decision threshold, reached a specificity of 0.58, sensitivity of 0.778, and accuracy of 68.7%, with 11/16 samples classified correctly (Figure 5).

## 4. Discussion

Accurate diagnosis is crucial for effective EC management, and there is a need for new preoperative molecular diagnostic tools [30,31]. We demonstrated that liquid biopsies could fulfil this niche in the future. To the best of our knowledge, this is the first study to explore potential of preoperative histological evaluation based on ctDNA profile in EC. Of importance, liquid biopsy analysis typically does not provide information on the type of tumor histology.

TEPs analysis so far has been successfully applied in NSCLC, ovarian cancer, and glioblastoma [2,23]. Our results suggest that TEPs transcriptional profile can also be used for distinguishing between EC patients and asymptomatic controls, as in other types of cancer, as earlier presented by Best et al. [2]. Nevertheless, as TEPs profiling usually indicates the tumor-site-of-origin localization, the discrimination between cancer cases and healthy subjects is effective, whereas distinguishing EC from BGC is less efficient. Therefore, imPlatelet classifier refinement should be considered with respect to the splice variants that specifically indicate EC.

In a traditional model, EC is divided into type 1 and type 2 cancer. Both types have mutations in certain genes: type 1 in PTEN, ARID1A, PIK3CA and KRAS, and type 2 in TP53. Currently, the histology evaluation relies on the uterine biopsy or uterine curettage [32]. This study points a potential of patient ctDNA mutational profile for preoperative assessment of EC histology. The classifier correctly categorized primary tumors of the test set, whereas ctDNA profile proved less informative, likely due to small sample size. In a similar attempt, Martinez-Garcia et al. used protein markers in fluid fraction of uterine aspirates for preoperative evaluation of tumor histology. Accuracy of their EC vs. non-EC differentiation was slightly lower than our TEPs classifier (96.0% vs. 100.0%), and the accuracy of serous vs. endometrioid EC was higher than our ctDNA classifier (99.0% vs. 68.7%) [33]. However, our method is more convenient for the patient, provides information on a larger number of molecular targets (71 genes instead of 29 proteins), and the applied machine learning makes the diagnosis more objective and time-efficient. This holds promise in the future for a clinical setting. On the other hand, our classifier was tested on a small number of ctDNA samples and should be validated on a larger group.

Several limitations of the study need be acknowledged. The first is a small number of liquid biopsies and the lack of matching between TEPs and ctDNA samples. Hence, the presented classifier accuracies should be intepreted with caution. However, for the training process, we have used all the asymptomatic controls (*N* = 204) and primary tumors (*N* = 519) we had had at our disposal at the moment of the analysis. Although we admit that enrolling more samples would likely result in the further refinement, we believe our results are meaningful and worth presenting despite the small liquid biopsy sample size. Additionally, we admit that imPlatelet classifier needs refinement to better distinguish BGC from EC. Further, under ideal conditions, BGC cohort should be limited to myoma, cyst, polyp and endometriosis cases, which are not considered pre-cancer conditions. The proposed approach is also expensive in a clinical setting, but the sequencing costs are expected to continue decreasing [34,35].

Random forest algorithm also needs refinement. The results of algorithm stability analysis (Appendix A) suggest that collecting more training data will result in significant improvements in the classification performance. These data should preferably be collected through liquid biopsies, since our analyses suggest that the data from biopsies and solid tumors do not follow the same distributions (Appendix A). To further test this hypothesis, more data are needed. We also observed that mutation frequencies in type-specific genes differed from the literature data [14,15]. For example, TP53 and ERBB2 gene mutations were found in both endometrioid and non-endometrioid EC types, and PTEN mutation frequency was lower than expected. This might be due to limitations of the used gene panel, or the fact that some mutations in ctDNA were below the limit of detection and thus remained uncalled. There is also a risk of false-negative results due to low volume of plasma yield and thus limited total number of available genome copies. Previous studies showed that tumor fraction of cfDNA varies between cancer types, and even metastatic tumors may demonstrate low amounts of ctDNA [26,36,37,38].

One of the advantages of applying machine learning is its flexibility. Recent publications provide evidence that type 1 and 2 EC classification should be replaced with ultramutated and non-mutated POLE gene discrimination [39,40,41,42]. Therefore, the division of endometrial cancer into type 1 and type 2 may shortly be considered obsolete. In the future, the developed random forest-based classifier can be adapted to optimize classification.

## 5. Conclusions

In conclusion, liquid biopsies show a great promise in EC. Our preliminary results show that both TEPs and ctDNA profiles constitute informative material for EC diagnosis and management. Further work, with a larger sample size and refined classifiers is warranted. In the future, the presented test might be strategically positioned as a screening tool. The interrogation of TEPs profile would aid initial EC diagnosis, particularly discrimination between healthy women and EC patients. Meanwhile, ctDNA analysis could serve as means of preoperative EC histology evaluation. This feasibility study shows that both TEPs and ctDNA profiles constitute informative material for EC diagnosis and management. Further work with a larger sample size and refined classifiers is warranted. In the future, both TEPs and ctDNA profiling can be considered in monitoring treatment response in EC patients.

## Figures and Tables

**Figure 1 cancers-13-05731-f001:**
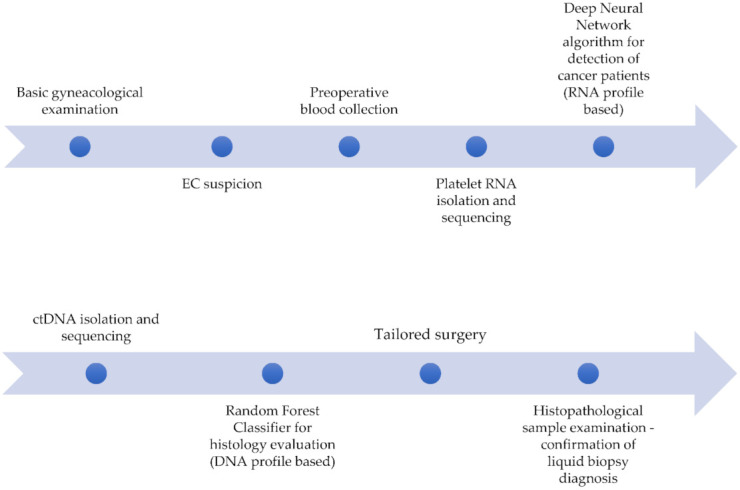
The concept of the future EC diagnosis and initial therapy planning based on the proposed experiment.

**Figure 2 cancers-13-05731-f002:**
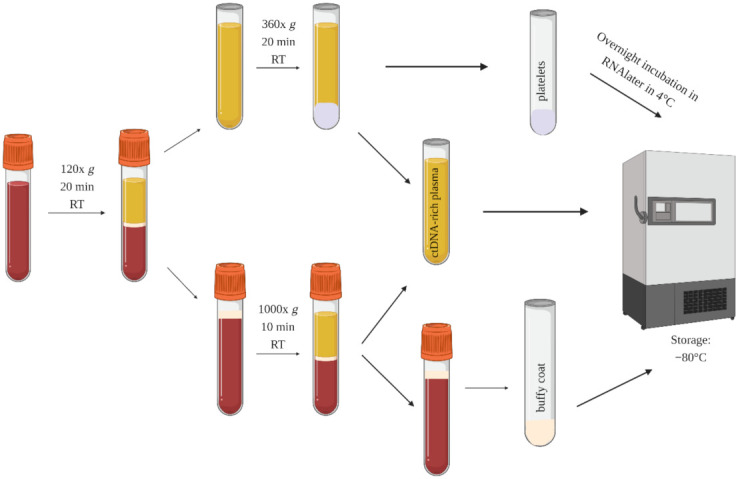
The blood processing protocol for the collection of platelets, ctDNA-rich plasma (somatic mutations), and buffy coat (germline mutations).

**Figure 3 cancers-13-05731-f003:**
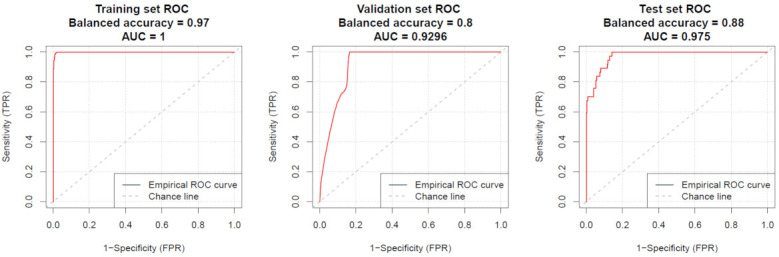
Receiver operating characteristic (ROC) curves representing the accuracy of imPlatelet classifier performance in the training, validation, and independent test set.

**Figure 4 cancers-13-05731-f004:**
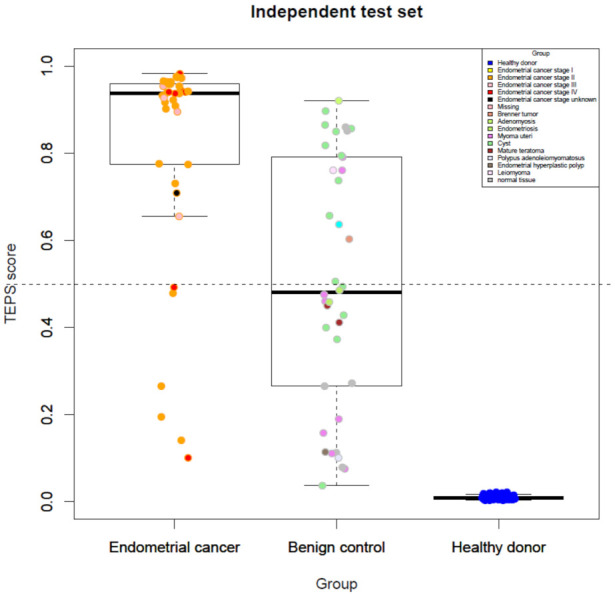
Probability scores for TEPs sequencing results in the test set.

**Figure 5 cancers-13-05731-f005:**
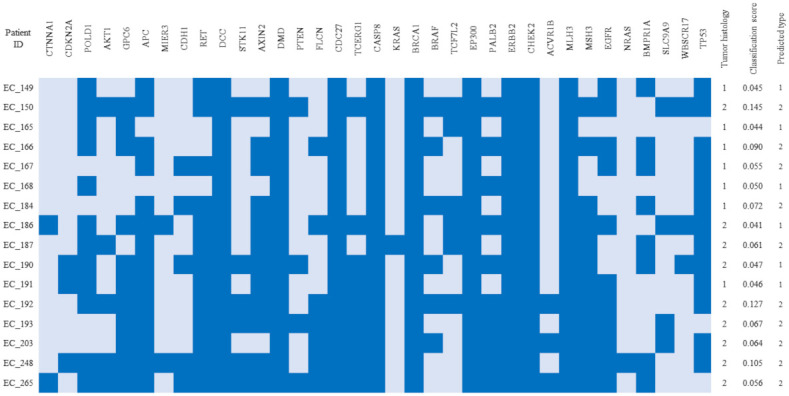
The summary of ctDNA sample classification compared with histology evaluation. The columns contain information regarding the genes that had nonsynonymus, stop gain or frameshift mutations in the corresponding samples (marked in dark blue). Tumor histology: 1–endometrioid, 2–non-endmetrioid.

**Table 1 cancers-13-05731-t001:** Clinicopathologic data of the study subjects: liquid biopsy of patients with EC, liquid biopsy of patients with benign gynecologic conditions (BGC, MUG cohort), liquid biopsy of healthy controls and EC primary tumors (GDC cohort). * BGC, benign gynecologic conditions, NA–not applicable.

Variable	Liquid Biopsy of EC Patients—MUG Cohort (*n* = 53)	Liquid Biopsy of BGC * Patients—MUG cohort (*n* = 38)	Liquid Biopsy of Healthy Donors—Dutch Cohort (*n* = 204)	Primary Tumors—GDC Cohort (*n* = 519)
Age				
<50	2 (3.8%)	19 (50.0%)	127 (62.3%)	45 (8.7%)
>50	50 (94.3%)	19 (50.0%)	77 (37.8%)	471 (90.8%)
Missing data	1 (1.9%)	0	0	3 (0.6%)
Histology				
Endometrioid	31 (58.5%)	NA	NA	389 (75.0%)
Non-endometrioid	22 (41.5%)	NA	NA	130 (25.1%)
Stage				
IA-IB	31 (58.5%)	NA	NA	326 (62.8%)
II	5 (9.4%)	NA	NA	51 (9.8%)
IIIA-IIIC	9 (17.0%)	NA	NA	116 (22.4%)
IVA-IVB	1 (2.0%)	NA	NA	26 (5.0%)
Missing data	7 (13.2%)	NA	NA	0
Grade				
1	7 (13.2%)	NA	NA	96 (18.5%)
2	24 (45.3%)	NA	NA	118 (22.7%)
3	16 (30.2%)	NA	NA	305 (58.8%)
Missing data	6 (11.3%)	NA	NA	0

## Data Availability

Raw data: Platelet expression data from healthy controls is available from GEO under accession number GSE89843. Platelet expression from benign gynecologic condition controls is available under GSE158508. Raw files with expression data of endometrial cancer patients are deposited at Gene Expression Omnibus, under the accession number GSE184904. Mutation data for ctDNA analysis is available under https://drive.google.com/file/d/1Qa4MaXEsNVpIktzzNLTcOZxjLMyzG8hp/view?usp=sharing (accessed on 30 September 2021). Normalized reads data is available at github: https://github.com/KrzysztofPastuszak/ImPlatelet_EC. Code availability: Code and docker image for ctDNA analysis is available under: https://drive.google.com/file/d/1Qa4MaXEsNVpIktzzNLTcOZxjLMyzG8hp/view?usp=sharing. Code for TEPs-based classifier is available under https://github.com/KrzysztofPastuszak/ImPlatelet_EC (accessed on 30 September 2021).

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
