# Peer review of "Diagnostic Accuracy of Liquid Biopsy in Endometrial Cancer"

_cancers, 2021, doi:10.3390/cancers13225731_

Round 1

Reviewer 1 Report

Liquid biopsy in patients with cancer is gaining more interests. The authors studied the utility of TEPs and ctDNA in identifying and distinguish between benign condition and endometrial cancer. Out of (I guess randomly) 297 selected patients only 53 patients had endometrial cancer and 40 had different benign uterine diseases. The rest of the patients were healthy and not known to have any uterine illnesses. The results are nicely showing that TEPs has a high value to distinguish between patients with uterine cancer and healthy patients. However, results do not have same high value in discriminating between patients with uterine cancer and other patients with benign GYN diseases. The authors used artificial intelligence to help them identifying RNA and DNA sequences. This part of deep neural network algorithm which was performed in other studies as well, is outside my expertise and I cannot comment on it.

Author may comment more on TEPS in the introduction and its origin.

Line 36 influence(s)

Line 209: liquid biopsy analysis typically does not provide information on the (type of) tumor histology

Line 261: Typo ThisOur feasibility study

Line 265: The interrogation of TEPs profile could would

Do the author think a step forward and think that the utility of TEPs and ctDNA may apply on monitoring of responses to treatment in patients with endometrial cancer?

Author Response

Dear Reviewer, 

Please find attached our response for your comments. 

We hope that it clarify your concerns. 

Reviewer 2 Report

The paper studies the utility of tumor educated platelets (TEPs) and circulating tumor DNA (ctDNA) for preoperative endometrial cancer diagnosis, including histology determination. Liquid Biopsy is the future of the targeted oncological treatment and the paper show for the first time a preoperative histological evaluation based on ctDNA profile in EC. The use of machine learning and artificial intelligence is a good solution to standardize the analysis of the data. 

In my opinion the sample size is too small to define the diagnostic accuracy of the reported methodologies, and the main novelty of the paper is supported by only 16 samples. Furthermore, it is not clear if the collected tissues correspond to the liquid biopsy.

The selection of the study population is not clear, statistical power of the study seem to be not calculated prior to the study. I do not agree to include in benign condition even pre-cancer conditions. 

Specific comments

Line 175 it is not clear if was collected only liquid biopsy as samples for the analysis or even the matched endometrial tissues. 

Line 176 -177 I do not agree to include in the control group pre-cancer condition as Atypical endometrial hyperplasia.

Line 178-179 I do not find a more detailed sample list in Table S4 e S5

 Line 197-198 liquid biopsy for ctDNA was matched with gene expression panel of endometrial tissue from the same patient? If affirmative the analysis was performed on preoperative biopsy or collected after demolition surgery? 

Line 230 -231 In my opinion the results of the study and the small size of the sample do not support this statement due to the lack of matching between TEPs and ctDNA samples.

Author Response

(The authors gave the same response as above.)
